# DNA Ligase 4 Contributes to Cell Proliferation against DNA-PK Inhibition in *MYCN*-Amplified Neuroblastoma IMR32 Cells

**DOI:** 10.3390/ijms24109012

**Published:** 2023-05-19

**Authors:** Kiyohiro Ando, Yusuke Suenaga, Takehiko Kamijo

**Affiliations:** 1Research Institute for Clinical Oncology, Saitama Cancer Center, Saitama 362-0806, Japan; tkamijo@saitama-pho.jp; 2Chiba Cancer Center Research Institute, Chiba 260-8717, Japan; ysuenaga@chiba-cc.jp

**Keywords:** neuroblastoma, MYCN, DNA-PK, DNA ligase 4

## Abstract

Identifying the vulnerability of altered DNA repair machinery that displays synthetic lethality with *MYCN* amplification is a therapeutic rationale in unfavourable neuroblastoma. However, none of the inhibitors for DNA repair proteins are established as standard therapy in neuroblastoma. Here, we investigated whether DNA-PK inhibitor (DNA-PKi) could inhibit the proliferation of spheroids derived from neuroblastomas of MYCN transgenic mice and *MYCN*-amplified neuroblastoma cell lines. DNA-PKi exhibited an inhibitory effect on the proliferation of *MYCN*-driven neuroblastoma spheroids, whereas variable sensitivity was observed in those cell lines. Among them, the accelerated proliferation of IMR32 cells was dependent on DNA ligase 4 (LIG4), which comprises the canonical non-homologous end-joining pathway of DNA repair. Notably, LIG4 was identified as one of the worst prognostic factors in patients with *MYCN*-amplified neuroblastomas. It may play complementary roles in DNA-PK deficiency, suggesting the therapeutic potential of LIG4 inhibition in combination with DNA-PKi for *MYCN*-amplified neuroblastomas to overcome resistance to multimodal therapy.

## 1. Introduction

Neuroblastomas are the most common extracranial solid tumours in early childhood and are often refractory or relapsed despite multimodality therapy. The International Neuroblastoma Risk Group classification system was recognised as a consensus for optimal treatment strategies [1]. In addition to the six potential prognostic factors, tumour stage, histology, *MYCN* amplification, age at diagnosis, the 11q aberration, and DNA ploidy, tumours with any type of segmental chromosome alterations characterised patients with a high risk of relapse [2]. Notably, the distribution analysis of DNA sequence elements and genomic landmarks in breakpoint intervals suggest that chromosome rearrangements in neuroblastomas are partly causally related via repair machinery of the DNA double-strand break (DSB) lesions, especially non-homologous end joining (NHEJ) [3].

Targeting components of the DNA repair machinery has long been considered as a treatment strategy for cancer, including neuroblastoma, because most cancers possess gene aberration of these components. Therefore, the synthetic interaction between BRCA1/2 mutation and poly (ADP-ribose) polymerase 1 (PARP) inhibitor was successfully translated for clinical application [4]. However, an effective inhibitor in the same category for treating patients with neuroblastoma has yet been approved for clinical use. Homologous recombination (HR) and NHEJ are the two major DSB repair pathways. HR is performed under the guidance of an intact homologous template DNA for high-fidelity repair, whereas NHEJ ligates DNA ends either directly or by processing microhomology. Therefore, NHEJ is an error-prone repair pathway; however, it theoretically functions as the predominant DSB repair pathway throughout the cell cycle [5]. NHEJ involves two major processes, a canonical NHEJ (c-NHEJ) and an alternative NHEJ (alt-NHEJ), which differ with respect to repair factors and end-joining fidelity. DNA ligases encoded by the human *LIG1*, *LIG3*, and *LIG4* genes belong to the nucleotidyltransferase family and perform the final step of NHEJ by sealing a single-stranded nick [6]. c-NHEJ is catalysed by enzymatic components, including polymerases of Pol μ and Pol λ, a nuclease complex of the Artemis−DNA-dependent kinase catalytic subunit (DNA-PKcs), and a ligase complex of the XLF−XRCC4−DNA Ligase IV (LIG4). Deletion of the murine *Lig4* gene leads to late embryonic lethality, presumably caused by neuronal cell death [7,8] and impaired lymphopoiesis owing to a defect in V(D)J recombination [9]. Furthermore, Lig4-deficient embryonic fibroblasts show hypersensitivity to ionising radiation [9]. Humans with inherited mutant alleles of *LIG4* exhibit various growth defects, namely, DNA ligase IV syndrome, which shares immunodeficiency as a common clinical symptom. In contrast, alt-NHEJ relies on PARP1 and DNA Ligase IIIα (LIG3) or DNA Ligase I (LIG1) by low fidelity, resulting in pathogenic chromosomal errors [10].

DNA-dependent kinase (DNA-PK) belongs to the phosphatidylinositol 3-kinase-related kinase family, together with the ataxia-telangiectasia mutated kinase (ATM) and the ATM and Rad3-related kinase (ATR). The DNA-PK catalytic subunit (DNA-PKcs) is encoded by the *PRKDC* gene. It forms a holoenzyme DNA-PK with the heterodimer regulatory subunits of Ku70 and Ku80 (henceforth, Ku) at DSB sites. Ku-deficient mice reportedly exhibit hypersensitivity to ionising radiation. Similarly, the severe combined immune deficiency mouse is characteristically deficient in DNA-PK activity and highly radiosensitive. Consequently, DNA-PK emerged as a therapeutic target for various cancers as a sensitiser for chemo- and radiotherapy that inflict DSBs [11]. *PRKDC* mRNA expression levels were enhanced in patients with neuroblastoma and correlated with a more advanced tumour stage and poor prognosis. The DNA-PKcs small-molecule inhibitor NU7026 synergistically radio-sensitised neuroblastoma cell lines [12,13]. The early-phase clinical trials of several DNA-PK inhibitors in combination with or without other chemotherapies and radiotherapy in adult patients with cancer are currently ongoing [14,15]. Notably, loss-of-function RNAi screening identified the *PRKDC* gene as a novel candidate for synthetic lethality in L-MYC-overexpressing lung fibroblasts [16]. As L-MYC and MYCN (also referred to as N-MYC) are MYC family members, it prompted us to investigate whether DNA-PKcs inhibitor (henceforth, DNA-PKi) could inhibit the cell growth of *MYCN*-amplified neuroblastoma. 

## 2. Results

### 2.1. Sensitivity to DNA-PKi Is Variable in MYCN-Amplified Neuroblastomas

To investigate the functional role of DNA-PK in MYCN-driven neuroblastoma, cell viability assays of neuroblastoma spheroids with or without treatment with DNA-PKi were performed. These neuroblastoma spheroids were derived from naturally occurring mouse abdominal and thoracic neuroblastomas from transgenic mice, in which the human *MYCN* gene was constitutively expressed under the control of the rat tyrosine hydroxylase promoter (TH-MYCN mouse [17]). Spheroids prepared from neuroblastomas of TH-MYCN mice were cultured using cancer-tissue-originated spheroid (CTOS) methods [18] to purify tumour cells (Figure 1A and Appendix A). We used NU7441 as a selective DNA-PKi [14]. The ATM inhibitor Ku55933 and the CHK1 inhibitor PF-477736 were also evaluated as references of inhibition for other DNA repair machinery, HR and the single-strand break repair, respectively [19]. The experimental concentration rages that adequately inhibit ATM [20,21] and CHK1 [19,22] were set as previously described. Cellular ATP levels were used to assess cell viability. As seen in Figure 1B, 96 h after treatment with each inhibitor, 2 μM DNA-PKi effectively inhibited the growth of the abdominal spheroids relative to ATMi and CHK1i. 

As a cell line is genetically variable owing to its dispersive chromosome aberrations, we investigated whether a differential response to DNA-PKi could be observed in several *MYCN*-amplified neuroblastomas. Increased concentrations of DNA-PKi revealed that SKN-BE and NBLS exhibited low sensitivity compared with that of IMR32, SMS-SAN, and CHP134 cells (Figure 1C). In addition, the inhibitory effect of NU7441 on cell growth at a comparable concentration has been reported in other types of cancer cell lines [23]. Furthermore, inhibition of cell growth was causally related to apoptosis in SMS-SAN and CHP134, and not in IMR32, as confirmed by the cleaved caspase 3 expression (Figure 1D). Thus, SK-N-BE, NBLS, and IMR32 cells might have the ability to protect, to some extent, their survival against DNA-PKi.

### 2.2. Increased Expression of DNA Ligase 4 (LIG4) Is Associated with the Worse Prognosis in Patients with MYCN-Amplified Neuroblastoma

To investigate the possible association between MYCN and DNA-PKcs or other enzymatic components of both c-NHEJ and alt-NHEJ, the correlation of mRNA expression levels of the two genes was analysed using the database R2, a Genomics Analysis and Visualization Platform. As shown in Figure 2A, the expression levels of *MYCN* and most of the components of NHEJ exhibited a positive correlation, whereas only *LIG4* expression had a significant inverse relationship with *MYCN* expression in the 88 neuroblastoma dataset (Versteeg-88-MAS5.0-u133p2). In contrast to the mutually exclusive expression profile of *LIG4* with the *MYCN* oncogene, the increased expression of *LIG4* was correlated with the unfavourable prognosis in the 66 and 92 MYCN-amplified neuroblastoma datasets (Tumour Neuroblastoma Kocak-649-custom-ag44kcwolf, raw *p* = 0.000056 and Bonferroni corrected *p* = 0.0028, Figure 2B; Tumour Neuroblastoma SEQC-498-RPM-seqcnb1, raw *p* = 0.0025 and Bonferroni corrected *p* = 0.194, Appendix A, respectively), suggesting that LIG4 could be involved in an oncogenic potential with relevance to MYCN.

### 2.3. LIG4-Ablated IMR32 Cells Exhibited a Delay in Cell Growth and Promoted Cell Susceptibility to DNA-PKi

As LIG4 acts as part of mainstream NHEJ, we investigated the possible implication of insensitivity to DNA-PKi in *MYCN*-amplified neuroblastomas. *LIG4* mRNA and protein expression levels were determined in SK-N-BE, NBLS, and IMR32 cells, which exhibited relatively low sensitivity to DNA-PKi. As shown in Figure 3A, *LIG4* mRNA was relatively upregulated in NBLS and IMR32 compared with that in SK-N-BE. In contrast, the expression of LIG4 protein increased in IMR32 relative to SK-N-BE and NBLS (Figure 3B). Therefore, we investigated whether the genetic disruption of *LIG4* could affect the cellular response to DNA-PKi in IMR32. To this end, we genetically engineered the ablation of LIG4 in IMR32 cells using the CRISPR/Cas-9 system and assessed cell growth with or without DNA-PKi. The antiproliferative effect was assessed by Alamar Blue dye (resazurin) reduction. A residual expression of LIG4 was observed in the genetically ablated clone by immunoblotting analysis, suggesting that LIG4 might be a lethal gene in IMR32 cells (Figure 3C). The LIG4-ablated IMR32 cells demonstrated significantly impaired cell growth compared with that of parental IMR32 cells (hereafter referred to as control cells), suggesting that LIG4 might be haploinsufficient and play a supporting role in MYCN-accelerated cell proliferation. Concordantly, in response to DNA-PKi, LIG4-ablated IMR32 cells exhibited a significant delay in cell growth relative to the control cells, resulting in decreased cell viability after treatment with DNA-PKi. (Figure 3C). Although the reproducibility of the results showing the antiproliferative effect of DNA-PKi in LIG4-ablated IMR32 cells was confirmed using an ATP assay, cells still retained their active metabolism, indicating that the cause of significant delay in cell growth might not be related to massive cell death (see Appendix A). Consistently, the siRNA-mediated knockdown of LIG4 in SK-N-BE and NBLS cells (the knockdown efficiency was around 70% and 30%, respectively) resulted in a slight decrease in cell proliferation (see Appendix A). Furthermore, the antiproliferative effect of DNA-PKi was potentiated in SK-N-BE cells with LIG4 knockdown, but not in NBLS cells (Appendix A), presumably due to differential dependency on a complementation mechanism such as alt-NHEJ. To determine if the combination effect of LIG4 deficiency was specific to DNA-PK inhibition, cell viability assays were performed in LIG4-ablated IMR32 cells and control cells in the presence or absence of CHK1i or ATMi. ATMi showed no apparent effect on LIG4-ablated IMR32 cells compared to control cells (see Appendix A). In sharp contrast to the effect on neuroblastoma spheroids, CHK1i was highly toxic in both LIG4-ablated IMR32 cells and the control cells at comparable levels (see Appendix A). This suggests that LIG4 may contribute to cell proliferation, to a lesser extent, against CHK1i and ATMi relative to DNA-PKi.

To gain mechanistic insights into the combination deficiency of DNA-PKcs and LIG4, immunoblotting was performed for representative markers of various cellular responses. Counterintuitively, although LIG4-ablated IMR32 cells exhibited a modest increase in cleaved caspase 3 without treatment, DNA-PKi treatment did not result in a substantial increase in cleaved caspase 3 (Figure 3D). PARP cleavage, an early marker of apoptosis, was observed similarly to cleaved caspase 3, but was more obvious. Notably, phosphorylated levels of p53 at Ser 15 were constitutively higher in LIG4-ablated IMR32 cells, indicating the activation of the p53 pathway. Thus, the delay in cell proliferation due to LIG4 deficiency may be attributed to the p53-related pathway. Interestingly, LIG4-ablated IMR32 cells exhibited a significant increase in p21, a downstream transcriptional target of p53 that leads to cell cycle arrest at G1 phase, whereas this increase was diminished in response to DNA-PKi. These findings suggest that a combination deficiency of LIG4 and DNA-PKcs leads to severe cell cycle delay due to the shift in the DNA repair pathway from NHEJ to HR, which requires newly synthesised sister chromatids, because NHEJ is a fast process that can be completed in approximately 30 min, while HR takes 7 h or longer to complete in a human cell [24]. Consistently, despite treatment, LIG4-ablated IMR32 cells exhibited a substantial increase in cyclin B expression, which is a marker of the G2 phase of the cell cycle (Figure 3D). In addition, a remarkable increase in phosphorylated ATM was observed in LIG4-ablated IMR32 cells, confirming an increased activity of DSB repair (Figure 3D). Collectively, these results indicated the therapeutic implications of LIG4 inhibition in combination with DNA-PKi for *MYCN*-amplified neuroblastomas.

## 3. Discussion

We observed that *MYCN*-driven neuroblastoma spheroids were relatively sensitive to DNA-Pki, which was consistent with the previous finding that L-MYC has synthetic lethal interactions with DNA-PKcs [16]. However, further in vitro experiments revealed that LIG4, the worse prognostic factors in *MYCN*-amplified neuroblastomas, might play complementary roles in DNA-PKcs deficiency. Considering the clinical heterogeneity of unfavourable neuroblastomas attributed to segmental chromosome alterations, additional gene alterations offering a survival advantage to *MYCN*-driven neuroblastoma are likely continually generated by NHEJ repair machinery. Substantial evidence revealed that the frequency of allelic loss at 11q23 was inversely related to *MYCN* amplification in neuroblastomas [25], a common region of deletion that harbours a series of proteins that play a central role in the DSB repair pathways such as ATM, MRE11, and H2AFX [25,26]. Notably, the function of DNA-PKcs in DNA end ligation is crucially regulated via phosphorylation by ATM concomitantly with auto-phosphorylation [27,28]. The ATM-dependent phosphorylation of DNA-PKcs is necessary for DNA-PKcs release for ligase reaction; however, the precise order of these sequential phosphorylation events is not fully understood [29]. These findings may explain why the chromosomal deletion of 11q rarely occurs coincidently with *MYCN* amplification. Collectively, we presume that *MYCN*-mediated neuroblastoma cell proliferation may largely depend on DNA-PKcs in coordination with ATM; however, if the activation of DNA-PKcs is impaired, LIG4 could complement their repair ability. Therefore, our results suggested that the increased dependency on the error-prone alt-NHEJ and time-consuming HR by the combined inhibition of DNA-PKcs and LIG4 may result in a severe delay in cell proliferation in IMR32, implicating the therapeutic potential of combined inhibition of these molecular targets (Figure 4). Moreover, c-NHEJ and alt-NHEJ were proposed as the targeting strategy using a PARP1i for treating neuroblastoma, indicating that alt-NHEJ also plays a complementary role in DNA repair for neuroblastoma cell survival [10]. The early phase of a clinical trial of PARPi for paediatric patients with refractory solid tumours is now underway and will address gene characteristics of the responders [30].

NU7441, used in this study, is a potent ATP-competitive DNA-PKi (IC_50_ vs. DNA-PK = 13 nM). Despite evidence supporting the role of NU7741 as a chemo- or radio-sensitiser in vitro and in vivo, further development could not be pursued because of the limited aqueous solubility and oral bioavailability [31]. Currently, clinical trials of selective DNA-PKi are followed by M3814 (also known as Nedisertib and Peposertib) for treating adult solid tumours and leukaemia in combination with or without other therapies [14], whereas a selective LIG4 inhibitor is not currently available. Therefore, exploitation of LIG4 inhibitors is desired, and efficient drug delivery may be required for avoiding predictive haematotoxicity.

## 4. Materials and Methods

### 4.1. Cell Culture and Inhibitors

We obtained the human neuroblastoma cell lines SKN-BE, NBLS, IMR32, SMS-SAN, and CHP134 from the American Type Culture Collection (Manassas, VA, USA) and the RIKEN Bioresource Cell Bank, Tohoku University Cell Resource Center (Miyagi, Japan) [19,32,33]. All cell lines were cultured in RPMI 1640 medium supplemented with 10% heat-inactivated FBS, 50 µg/mL penicillin, and 50 µg/mL streptomycin (Thermo Fisher Scientific, Waltham, MA, USA). We purchased NU7441, a selective inhibitor of DNA-PKcs, from Sigma Aldrich (St. Louis, MO, USA).

### 4.2. Spheroid Culture

A number of 129x1/SvJ-Tg TH-MYCN mice were originally obtained by Dr. William Weiss (University of California, San Francisco) [17]. Naturally occurring neuroblastomas in a male MYCN homozygote (Tg/Tg) mouse at 45 days of age were used for generation of neuroblastoma spheroids [34]. After culturing by cancer-tissue-originated spheroid (CTOS) methods [18], the spheroids were dissociated into single cells by treating with TrypLE Select (×10) (Life Technologies, Carlsbad, CA, USA) for 30 min at 37 °C, and used for cell viability assay in vitro.

### 4.3. Cell Viability Assays

Cells dissociated from CTOSs were seeded in 96-well plates (PrimeSurface 96V, Sumitomo Bakelite, Tokyo, Japan) at 2000 cells/well for 24 h. After incubation, in presence or absence of each inhibitor for 96 h, cell viability was quantified using the KATAMARI ATP assay Kit Ver. 2.1 (Tokyo Bnet, Tokyo, Japan) according to the manufacturer’s instructions. Briefly, 100 µL of the ATP detection solution was added to each well, the plates were incubated for 10 min, and the chemiluminescence of each well was measured using a ARVOX3 micro plate reader system (Perkin-Elmer, Waltham, MA, USA). For *MYCN*-amplified cell lines, cells were diluted to 10,000 cells/100 µL and seeded into each well of a 96-well plate. After incubation for the indicated times and conditions, 10 µL alamarBlue Cell Viability Reagent (Invitrogen, Carlsbad, CA, USA) was added to the cells for 2 h, and the fluorescence intensity was measured using an infinite F200 PRO plate reader (TECAN, Männedorf, Zurich, Switzerland).

### 4.4. Quantitative Reverse-Transcription PCR of Gene Expression

Total RNA was isolated using a RNeasy mini kit (Qiagen, Hilden, Germany) and reverse-transcribed using random primers and ReverTra Ace (TOYOBO, Osaka, Japan). Relative quantification of mRNA expression was performed using the StepOnePlus Real-Time PCR System with the indicated TaqMan probes and TaqMan Fast advanced PCR Master Mix (Applied Biosystems, Waltham, MA, USA). *LIG4* and *ACTB* TaqMan probes (TaqMan Gene Expression Assay, Hs01866071_u1 and Hs01060665_g1, respectively) were purchased from Applied Biosystems.

### 4.5. Immunoblotting Analysis

Immunoblotting analysis was performed as described previously [19]. Whole-cell extracts (50–100 µg protein) were prepared and resolved on SDS-PAGE (XCell SureLock Mini-Cell and NuPAGE Bis-Tris Precast Gel, Thermo Fisher Scientific). Gels were transferred to nitrocellulose membranes (Bio-Rad, Hercules, CA, USA), blocked, and incubated with primary antibodies for 16 h. Membranes were washed and incubated with HRP-coupled secondary antibodies. The proteins were detected via chemiluminescence using the ECL Select Western Blotting Detection Reagent (Cytiva, Tokyo, Japan). Chemiluminescence measurements were performed using an ImageQuant LAS4000 system (FUJIFILM, Tokyo, Japan). The following antibodies were used: anti-DNA-PKcs (E6U3A), anti-caspase-3, anti-DNA Ligase IV (D5N5N), anti-ATM antibody (D2M2), anti-phospho-ATM-Ser1981 (D25E5), anti-PARP, anti-phospho-p53-Ser 15, anti-p21 (12D1) antibody (Cell Signaling Technology, Danvers, MA, USA; 1:1000), anti-β-Actin (20–33) (Sigma-Aldrich; 1:4000), anti-cyclin B (BD Transduction Laboratories, Franklin Lakes NJ, USA; 1:1000), anti-tubulin (B-5-1-2) (Sigma-Aldrich; 1:4000), anti-N-Myc (B8.4B), anti-p53 (DO-1) (Santa Cruz Biotechnology; 1:500, Dallas, TX, USA), and horseradish peroxidase-coupled anti-rabbit or anti-mouse secondary antibodies (Cell Signaling Technology; 1:2000).

### 4.6. Immunoblotting Analysis

TrueCut Cas9 Protein v2 (Invitrogen) and TrueGuide Synthetic gRNA targeting the *LIG4* gene locus (CRISPR659021_SGM, Invitrogen) were co-transfected into cells using Lipofectamine CRISPRMAX (Invitrogen) to ablate the DNA ligase 4 (*LIG4*) gene. Control cells were transfected with TrueCut Cas9 Protein v2 alone. The cells were incubated for 72 h, and LIG4 knockout single clones were established using limiting dilution.

### 4.7. Immunoblotting Analysis

Statistical differences were analysed using Mac ToukeiKaiseki version 3.0 (ESUMI Co., Ltd., Tokyo, Japan), and two-sided *p*-values < 0.01 were defined as statistically significant.

## Figures and Tables

**Figure 1 ijms-24-09012-f001:**
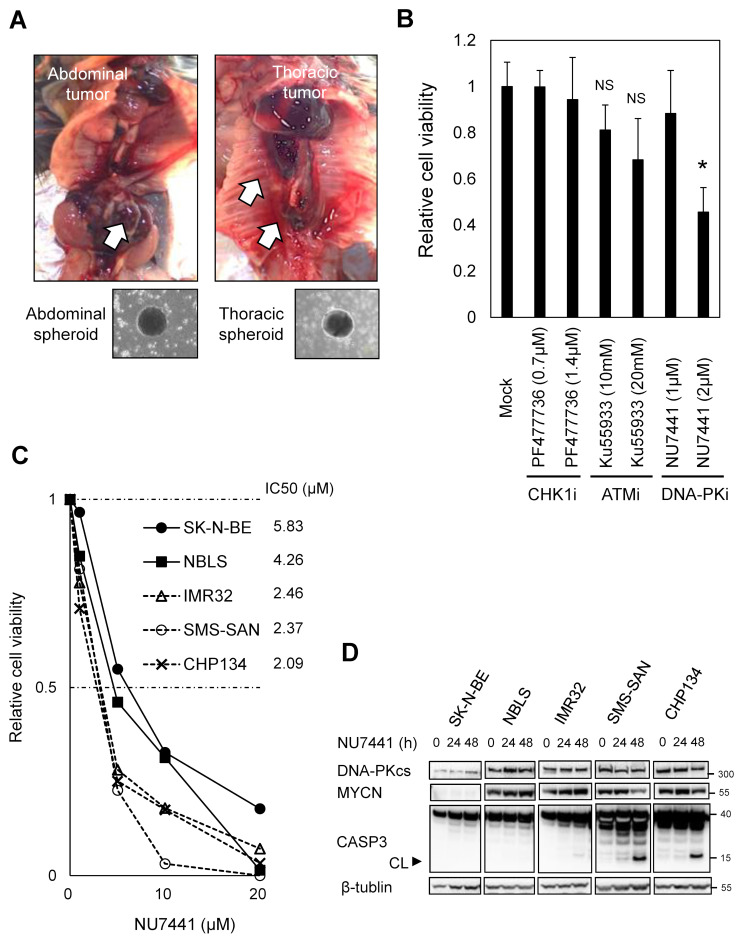
DNA-PK inhibitor (DNA-PKi) inhibits proliferation in *MYCN*-driven neuroblastoma spheroids and, to a lesser extent, in some *MYCN*-amplified neuroblastomas. (**A**) Neuroblastoma spheroids naturally occurring in TH-MYCN transgenic mice. White arrows denote abdominal and thoracic tumours. The lower panel presents the morphology of spheroids derived from each tumour sight. (**B**) Cell viability assays of abdominal neuroblastoma spheroids treated with DMSO (Mock), PF-477736 (CHK1 inhibitor, CHK1i), Ku55933 (ATM inhibitor, ATMi), and NU7441 (DNA-PKi) at the indicated concentrations for 96 h. (**C**) Cell viability assays of the indicated *MYCN*-amplified neuroblastoma cell lines treated with NU7441 at the indicated concentrations for 72 h. IC_50_ values for cell proliferation in each cell line are shown. (**D**) Immunoblotting analyses of the indicated proteins in MYCN-amplified cell lines after treatment with 3 µM DNA-PKi at no treatment (0 h) or indicated concentration for 24 h and 72 h. CL, cleaved caspase 3. Statistical significance is presented as the mean ± standard deviation (SD) of triplicates. * *p* < 0.01. NS, not significant.

**Figure 2 ijms-24-09012-f002:**
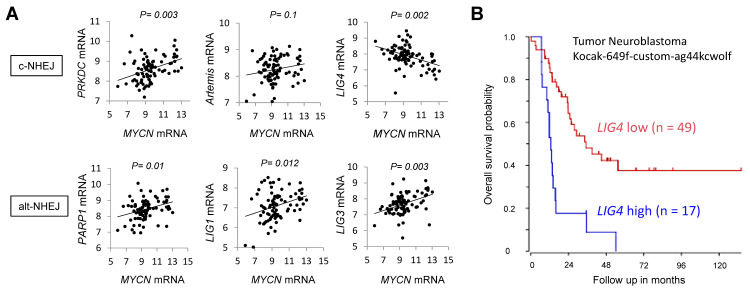
An mRNA expression of *DNA ligase 4 (LIG4)* is inversely related to that of *MYCN* and associated with poor prognosis in *MYCN*-amplified neuroblastomas. (**A**) Co-expression analyses between the indicated components of NHEJ and *MYCN* in 88 patients. Representative *p*-values were calculated using Pearson’s correlation coefficient test. (**B**) Kaplan–Meier survival curves of 66 patients with *MYCN*-amplified neuroblastoma stratified by high or low *LIG4* mRNA expression. c-NHEJ; canonical NHEJ, alt-NHEJ; an alternative NHEJ.

**Figure 3 ijms-24-09012-f003:**
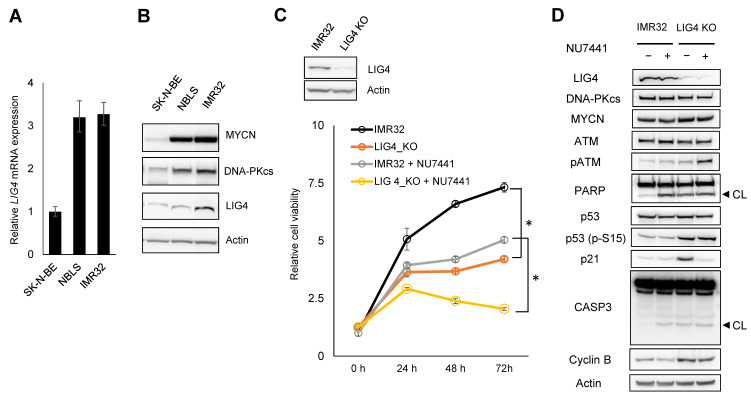
DNA ligase 4 (LIG4)-ablated IMR32 cells significantly impaired cell proliferation without inducing massive apoptosis. (**A**) Quantitative RT-PCR analysis, and (**B**) immunoblot analysis of LIG4 at basal levels in the DNA-PKcs inhibitor (NU7441) insensitive cell lines. (**C**) Immunoblotting (upper panel) and cell viability assays (lower panel) in IMR32 cells, in which LIG4 is ablated or not (LIG4_KO or IMR32, respectively) after treatment with or without 3 µM NU7441 for the indicated time course. Relative viable cell numbers are normalised by those of corresponding untreated cells. Statistical significance is presented as the mean ± standard deviation (SD) of triplicates. * *p* < 0.01. (**D**) Immunoblot analysis of the indicated protein in IMR32 and LIG4_KO cells after treatment with or without 3 µM NU7441 for 72 h. CL; cleaved form.

**Figure 4 ijms-24-09012-f004:**
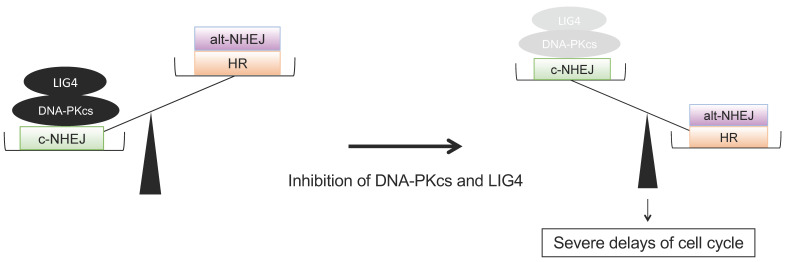
Schematic representation of the possible impact of the combined inhibition of DNA-PKcs and LIG4 on the significant delay of cell proliferation in IMR32. c-NHEJ; canonical NHEJ, alt-NHEJ; an alternative NHEJ. HR; homologous recombination.

## Data Availability

Not applicable.

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
