# Peer review of "DNA Ligase 4 Contributes to Cell Proliferation against DNA-PK Inhibition in MYCN-Amplified Neuroblastoma IMR32 Cells"

_ijms, 2023, doi:10.3390/ijms24109012_

Round 1
Reviewer 1 Report
In the manuscript "DNA ligase 4 complements cell proliferation against DNA-PK inhibition in MYCN-amplified neuroblastoma IMR32 cells", the authors provide evidence that the double loss of LIG4 and DNAPK is detrimental to cell growth in MYCN over expressing cells.
This manuscript is easy to read and of potential interest to general audience.
In general, this manuscript is not very novel as it has been established that the double loss of LIG4 and other canonical end-joining factors such as Ku70 and Ku80 is detrimental to cell survival, with this study only adding the concept of MYC expression.
This manuscript also appears very preliminary in nature. The authors appear to make conclusion from data collected only from the IMR32 cell line. It would benefit the paper if the authors used both the SK-N-BE and the NBLS cell lines as a comparison to the IMR32 cells.
While the authors demonstrate that loss of LIG4 and DNAPK inhibitor results in decreased cell viability, it would benefit manuscript to examine if this observation is specific to just the DNAPK inhibitor or does this observation also occur with inhibition of ATM and CHK1. While the authors demonstrated that the ATM and CHK1 inhibitor by themselves have no effect, they did not rule out the idea that there is a synthetic lethal event that can occur between the inhibitors and LIG4 loss.
There is no problem with how manuscript is written.
Reviewer 2 Report

This manuscript is well-written in English.
Round 2
Reviewer 1 Report
The authors have responded to all the comments made by this reviewer in the previous review. The current manuscript is significantly improved from the previous version.